# The Specificity of Motor Learning Tasks Determines the Kind of Skating Skill Development in Older School-Age Children

**DOI:** 10.3390/sports8090126

**Published:** 2020-09-14

**Authors:** Dominik Novak, Adam Tomasek, Patrycja Lipinska, Petr Stastny

**Affiliations:** 1Faculty of Physical Education and Sport, Charles University, 16252 Prague, Czech Republic; adam.tomasek6@seznam.cz; 2Institute of Physical Education, Kazimierz Wielki University in Bydgoszcz, 85-064 Bydgoszcz, Poland; patlipka@gmail.com

**Keywords:** motor learning, change of direction speed, training of partial skating, ice hockey, youth, agility, ice hockey skating

## Abstract

The specificity of motor learning tasks for skating development in older school-age children has not been sufficiently explored. The main objective was to compare the effects of training programs using change-of-direction (COD) speed exercises and partial skating task (SeqT) training on speed and agility performance in U12 ice hockey players. Thirteen young ice hockey males (13 ± 0.35 years, 41.92 ± 9.76 kg, 152.23 ± 9.41 cm) underwent three straight speed (4 and 30 m with and without a puck) and agility testing sessions before and after six weeks of COD training and then after a six-week intervention involving partial skating task (SeqT) training. The statistics were performed using magnitude-based decision (MBD) analysis to calculate the probability of the performance change achieved by the interventions. The MBD analysis showed that COD training had a large effect (11.7 ± 2.4% time decrease) on skating start improvement (straight sprint 4 m) and a small effect (−2.2 ± 2.4%) on improvement in agility with a puck. Partial skating task (SeqT) training had a large effect (5.4 ± 2.5%) on the improvement of the 30-m sprint with a puck and moderate effect on agility without a puck (1.9 ± 0.9%) and likely improved the 30-m sprint without a puck (2.6 ± 1.3%). COD training on the ice improves short starts and agility with a puck, while partial skating tasks (SeqT) target longer 30-m sprints and agility without a puck. Therefore, both types of training should be applied in accordance with motor learning tasks specific to current training needs.

## 1. Introduction

Skating is an elementary movement in ice hockey that makes players fast for straights, with speeds of more than 40 kph, and quick for changes in direction (CODs) [1,2]. Therefore, the skating skills represented by sprints, cutting maneuvers, turns, weave agility, breaks, hits and acceleration [3] represent the ability to achieve success in ice hockey [4,5]. Although ice hockey players skate daily during most of the season, specific skating interventions are necessary to keep skating skills and agility fully automatic during high-intensity effort [6]. Since skating is not a natural fundamental motor skill, its trainability is probably lower than that of other movements, such as jumps, and a specific approach might be needed to learn skating tasks.

Ice hockey skating skills are realized by linear skating, various types of CODs and accelerations and stick handling and checking [5,7]. These skills are realized in ice hockey matches and have to be performed with high motor control quality and automatization; for example, ice hockey players spend 39% of their total time on ice gliding on two feet, and the rest of their time is spent performing other movements (gliding, crossovers, backward skating, etc.). For this reason, it is important to develop transitions between linear skating and CODs [7,8]. Learning skills in ice hockey is a long-term process, where the aim is to develop highly automatized skills, which might be used intuitively. Beyond this level, skills are learned as whole COD tasks or by partial sequences of movements (SeqTs) to develop detailed skating elements. Between 11 and 13 years of age, players typically undergo dramatic anthropometric changes, cognitive emotional capabilities and strength, which raises the question of what learning approach should be preferred to improve ice hockey skating skills [9].

CODs and SeqTs are methods of motor learning that should be employed in learning processes [10,11] to improve ice hockey skating. The main difference is that CODs are based on already learned skating elements in sequences with the aim of synchronization of technical skills, straight skating speed, lower limb muscle qualities and anthropometry factors [12,13,14]. On the other hand, SeqT are based on the method of improving skills in parts, where the elements of skating are developed by partial skating movements (e.g., sliding, using the inner edge or outer edge of the blade); therefore, its effect is based on the precise development of elementary skating skills. In terms of skills development, the specific COD training has been reported for high effectiveness even in elite athletes, while there is lack of research of SeqT effects [15].

Since there is a lack of information on the specifics of trainability in skating, the main objective was to compare training programs using exercises for change of direction (COD) speed and sequence training of partial skating tasks (SeqT) on speed and agility performance in U12 ice hockey players. Under this aim, we hypothesized that COD training would have better results for any skating performance than partial skating task training.

## 2. Materials and Methods

### 2.1. Experimental Approach to the Problem

Thirteen sixth-grade ice hockey players from the same ice hockey club participated in this study. The study included three measurements. The first test measurement was the entry one (pre), the second follows after the intervention from the training of changes in directional speed (post-test 1), and the third test measurement was performed after the partial skating task intervention (post-test 2). All participants trained their change of direction speed for the first six weeks of training. After this training period, the participants rested for two weeks. After the first stimulation and rest, a program for partial skating was performed for another six weeks (Figure 1). Participants trained together for 12 weeks, three times per week for 20 min in the first part of the training unit, after 5 min of individual warm-up, which included three lower limb static stretching exercises, 20 short sprints and two minutes of stickhandling and individual agility drills. The obtained data were recorded in charts and statistically processed by the “a scale of magnitudes” method. This method enables the calculation of the scale of differences in the means and second of the probability of change being caused by the intervention. The entire experiment was performed during a competition season, where first measurement was performed on the second of December 2018, and the last measurement was performed on the third of March 2019. The training intervention was established by head coach (author A.T). in cooperation with research team D.M., P.S., P.L. containing exercise physiologist, biomechanics specialist and sport science experts.

### 2.2. Subjects

All participants (*n* = 13,13 ± 0.35 years, height 152.23 ± 9.41 cm, body mass 41.92 ± 9.76 kg, BMI 17.91 ± 2.14, Table 1) were male players in the sixth-school grade from the same ice hockey club. Five of them were defenders, and eight were forwards. They played in the highest youth league and participated in the same training program. The research and informed consent forms were approved by the institutional ethics committee of the Charles University Faculty of Physical Education and Sport in accordance with the ethical standards of the Helsinki Declaration of 2013, and signed informed consent forms were obtained from the parents of all players who participated in this study.

### 2.3. Regular Training Program

The participants’ regular in-season program consisted of weekly cycles, which included one or two competitive matches per week and four regular 60-min ice hockey training sessions per week. Regular ice hockey training sessions were focused on individual game activities, skating conditioning, individual offensive and defensive tactics, individual offensive activities and game strategies. We considered this regular training program, which is not targeted at providing additional adaptive stimuli for agility and skating skills, as the players’ long-term routine. During the first intervention (training of the change of direction speed), they had 23 training sessions, 18 stimulations and 9 matches. In the second intervention (partial skating training), they had 25 trainings, 18 stimulations and 4 matches.

### 2.4. Training of the Change of Direction (COD) Speed

We used change of direction speed trainings for developing ice hockey skills. The time of stimulation was 6 weeks, three times per week for 20 min during the first part of training [6]. Participants trained one exercise for 10–20 s, the minimum number of repetitions was 2 times, and the rest interval between repetitions was 1:3–5. During the first training of the week they did exercises no. 1 and no. 3 (Appendix B, Figure A1 and Figure A3), during the second training of week, they did exercise no. 2 (Appendix B, Figure A3), and during the last training of week, they practiced COD exercises with passing and shooting.

### 2.5. Training of Partial Tasks (SeqTs)

Partial skating training tasks are often used in youth ice hockey players so that difficult exercises or skills can be partitioned. During the program we used training focused on using different edges of the skate during skating tasks, like using inner and outer edge of blade for cutting maneuvers (Appendix C, Figure A4, Figure A5, Figure A6 and Figure A7). The players were using alternatively inner and outer edge of blade for forward and backward turns, push-offs and sliding to skate straight or change the skating direction. The time of stimulation was 6 weeks, three times per week for 20 min during the first part of training. Participants trained one exercise for 10–20 s, the minimum number of repetitions was 2 times, and the rest interval between repetitions was 1:3–5.

### 2.6. On-Ice Test Measurements

All tests were performed with photocells (Brower Timing System, Utah, USA). The starting photocells in all tests were placed 15 cm above the ice, and the finish sensors were placed 108 cm above the ice. [3] If someone fell or lost a puck during a measurement, the trial was restarted. Players self-started behind the timing gate with his stick over the gate. Test no. 1, 2, 3, 4 and 5 were used as skill tests at the Winter Youth Olympic Games 2010 by the International Ice Hockey Federation (IIHF) [6]. Test no. 6, 7 and 8 are different because the player sprints for only 4 m, and their measurement is taken at 6.1 m [16,17].

#### 2.6.1. Agility without Puck and with the Puck

The player started at the start/finish line without a puck (Figure 2. Test no. 1). The player skated forward to the far right pylon, pivoted forward to backward and skated backwards to the lower right pylon. Then, he pivoted from backward to forward, skated to the far left pylon, pivoted again forward to backward, and then skated to the lower left pylon. There were no backward-to-forward pivots; the player skated forward to the opposite side of the circle from the start/finish line. Then, the player stopped and skated forward back to the finish line. This was a timed drill; the time began when the player started moving and ended when he crossed the start/finish line (Figure 2. Test no. 1). The same track was performed with the puck for the Test no. 2 (Figure 2. Test no. 2). If the player lost the puck or fell during test no. 2, he was given a second chance.

#### 2.6.2. Straight 30-m Forward and Backward Sprint with and without a Puck

The player started forward on the time gate line without a puck for test no. 3 and with a puck for the test no. 4. He skated as fast as possible for 30 m straight ahead towards the finish line (Figure 2, Test no. 3 and 4). For backward skating, the player started backward on the time gate line without a puck for the test 5. He skated backward as fast as possible towards the finish line (Figure 2. Test no. 5).

#### 2.6.3. Straight 4-m Forward and Backward Sprint Speed without a Puck

The player started forward on the time gate line without the puck for the test no. 6 (Figure 2. Test no. 6) and with a puck for Test no. 7 (Figure 2. Test no. 7). He skated as fast as possible straight ahead towards the 4 m finish line. For the test no. 8, the player started backward on the time gate line without a puck and skated as fast as possible backward towards the 4 m finish line (Figure 2. Test no. 8).

### 2.7. Statistical Analyses

The comparison of the effects of the two interventions was performed using magnitude-based decision (MBD) analysis [18,19,20]. Changes in the post 1 and post 2 measurements between groups after each 6-week intervention were analyzed using a spreadsheet for the post-test analysis only. Threshold values for assessing the magnitude of the standardized effects were 0.20, 0.60, 1.2 and 2.0 for small, moderate, large and very large effects, respectively. The uncertainty of each effect was expressed as 90% confidence limits and as probabilities that the true value of the effect was beneficial, trivial or harmful. These probabilities were used to make a qualitative probabilistic clinical inference about the true effect [21,22]; if the effects were deemed clinically clear, then they were expressed as the chance of the true effect being trivial, beneficial or harmful using the following scale: 25–75%, possibly; 75–95%, likely; 95–99.5%, very likely; >99.5%, most likely.

## 3. Results

All 13 players participated in all training and testing sessions, where there was no injury during the study. The MBD analysis showed that COD training had a large or very large beneficial effect on skating start (straight 4-m sprint) with a high probability and a small likely beneficial effect on agility with a puck (Table 2, Appendix A). Partial skating task training (SeqT) has a beneficial effect on long sprints (30 m) with and without a puck and a moderate most-likely effect on agility without a puck (Table 2, Appendix A).

## 4. Discussion

This study showed that both methods (COD and SeqT) are beneficial for agility and straight sprints with and without a puck but that they were effective for different movement tasks. COD training improves short starts (straight speed for 4-m backward and forward) and agility with a puck. However, COD training did not improve cyclic movements such as straight sprints for longer distances, which is in agreement with a previous study [6]. SeqT training is more beneficial for straight speed during longer distances (30 m) but not short sprints and hybrid skills such as agility. Both training methods should be applied in accordance with specific motor learning tasks and current training needs.

Motor learning is a long-term progressive process that is based on early learning phases, and its generalization, learning transfer and learning stimulus overlap [10]. The participants in this study were already at the training level where skating skills are automatized, but they need to progressively develop more precise, fast and difficult tasks. Therefore, the finding that both COD and SeqT training can target specific skating skill development in bantams has a notable implication for practical use, especially if coaches can determine which skating task is lacking in terms of development. The motor learning processes present in COD and SeqT training are based on a certain amount of variability when repeating a key motor task, which has been shown to be a functional strategy for achieving high movement precision. Although the COD and SeqT trainings involved different learned movement patterns, they included the same basic strategy of motor control and thus produced specific learning results [3].

In this work, most of the tests were proposed by the IIHF to assess the quality of skating [23]. The results of individual tests can be compared with the results recorded between 2006 and 2015 in the Canada Hockey database including results of Ontario Hockey league (OHL) or with the results of other studies. However, comparison of the results is not entirely appropriate because players in other studies are usually older and more experienced than those in the group tested in this study, age in Canadian categories are 11–12 years for PeeWee, 13–14 years for Bantam, 15–17 years for Midget and between 15–20 years the Juniors [24]. 

In test no. 1 (agility without a puck), the average time of onset in our last measurement was 15.69 s, while Canadian junior players U17, it was 12.53 s and OHL players 11.43 s [25]. Our players achieved an average result of 17.27 in test no. 2 (agility with a puck) during the third measurement, which is similar to 16.43 ± 2.22 s reported in mix group of Canadian PeeWee and Juniors (age 14.6 ± 2.1 years; playing experience 8.9 ± 3.1 years) [26]. These differences in agility are understandable in relation to our players age and skating experiences.

In a 30-m straight sprint without a puck (Test no. 3), our players were slower (5.98 s) than Juniors (age 16.3 ± 1.7; playing experience 10.3 ± 3.0) for 35-m sprint without a puck (5.14 ± 0.21 s) [3] and slower than Norwegian Midgets (age 16.4 ± 0.6 years) with average measured time 5.28 ± 0.21 s [27]. An even bigger difference was found in 30-m straight sprint with a puck (In test no. 4), where measured time 6.4 ± 1.1 s in this study was slower than Canadian Juniors (U17) with average result 4.47 s, and OHL players averaged 4.08 s [25]. Similarly, backwards skating 30-m without a puck (Test no. 5), in our tested group was 7.63 s, which is slower than the 6.16 ± 0.31 s reported in Norwegian Midgets (age 15.8 ± 0.9 years) for a longer distance—36 m [27].

The shortest test for a 4-m sprint used in this study had average performance at initial measurement 1.9 ± 0.1 s, which can be compared to Junior players from Division III (age 20.5 ± 1.4 years) who averaged 1.34 ± 0.3 s [16]; however, this study used a distance of 6.1 m. Woman‘s hockey players (age 12.18 ± 2.05 years; playing experience 4.68 ± 2.69 years) reached in 6.1 m sprint speed 1.63 ± 0.12 s [17]. This shows that study participants had lower initial level of acceleration, which was improved by COD training by 0.23 s (Appendix A). The improvement after COD was higher than previous improvement in on- and off-ice agility training programs in Juniors (age 14.8 ± 0.45; playing experience 9.07 ± 0.75 years) [6]; their time was 1.33 ± 0.28 s after on-ice training, and 1.31 ± 0.11 s after off-ice agility training [6]. However, this study used a test distance of 6.1 m. Test no. 7 and 8 are not included in other studies, which disable the studies comparison.

Both training methods (COD and SeqT) resulted in an improvement in the test battery, although COD training was probably more effective in terms of short sprints and agility, and SeqT training was more effective for long-distance skating. We consider the improvements in both programs to be significant in terms of performance and recommend both training methods for training during the ice hockey season.

The main limitation of this study is the absence of cross-sectional design which, in our case, can favor the first COD intervention. However, since the players in the study are interchanging both types of training during regular season, we can neglect large learning effect from the first intervention. Moreover, we can highlight the specificity of motor learning task, which can specify the use of both methods when coaches observe deficits in agility or straight speed. Another limit is the sample size, which is borderline; however, the sample size allowed a high level of control of the whole training process. Since there are differences in the skating technique and overall physical conditions between genders and performance levels, our results should be interpreted specifically to the male players at the reported age and experience [1,28,29,30]. 

## 5. Conclusions

There is evidence from the results that COD training has better effects for forward and backward starts and partial skating training has better effects for straight speed; however, it could be applied the other way around as well. Based on the results, we can say that partial skating is not effective for skating starts. This knowledge can be used by ice hockey players and trainers to create training plans and to effectively combine COD and partial task conditioning. An understanding of COD and partial task skating can improve skating performance and direct our focus in ice hockey training. Therefore, we recommend the interchange of COD and partial skating training during the ice hockey season.

## Figures and Tables

**Figure 1 sports-08-00126-f001:**
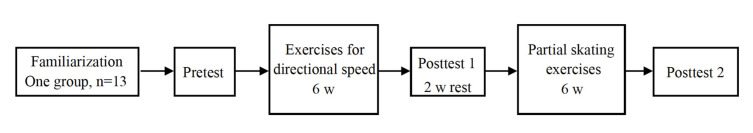
Methodological design of the test and exercise protocols.

**Figure 2 sports-08-00126-f002:**
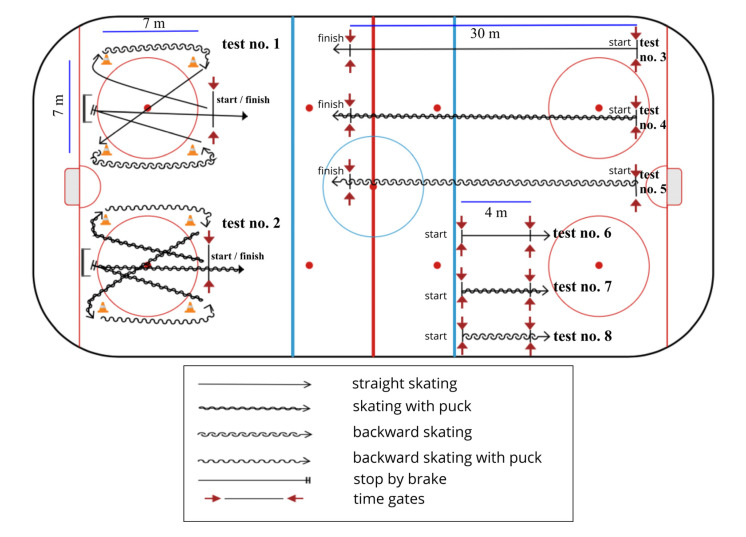
Graphical track and the distances of the on-ice tests and legend description. Test no.1: Agility without a puck; test no. 2: Agility with a puck; test no. 3: straight 30-m sprint without a puck; test no. 4: straight 30-m sprint with a puck; test no. 5: backward skating for 30 m without a puck; test no. 6: straight 4-m sprint speed without a puck; test no. 7: straight 4-m sprint speed with a puck; test no. 8: backward 4-m sprint speed without a puck.

**Table 1 sports-08-00126-t001:** Basic characteristics of subjects.

Subject	Height	Weight (kg)	BMI	Post
1	152	38	16.45	Defender
2	146	36	16.89	Forward
3	166	62	22.5	Forward
4	133	35	19.79	Defender
5	161	48	18.52	Defender
6	155	43	17.9	Defender
7	160	50	19.53	Forward
8	154	37	15.6	Forward
9	138	29	15.23	Defender
10	160	39	15.23	Forward
11	147	37	17.12	Forward
12	146	37	17.36	Forward
13	161	54	20.83	Forward
Mean ± SD	152.23 ± 9.41	41.92 ± 8.76	17.91 ± 2.14	

**Table 2 sports-08-00126-t002:** Baseline, percent change and magnitude-based decision (inference) after two different interventions.

Test	Baseline Mean ± SD	After COD		After Partial Task Training	
		Mean ± 90% CL	Effect	Mean ± 90% CL	Effect
Agility without a puck (s)	16 ± 0.4	−0.4%; ± 1.3%	unclear	−1.9%; ± 0.9%	Moderate ****
Agility with a puck (s)	18 ± 1.0	−2.2%; ± 2.4%	Small **	−1.9%; ± 2.1%	small *
Straight 30-m sprint without a puck (s)	6.1 ± 0.4	0.3%; ± 2.0%	unclear	−2.6%; ± 1.3%	Moderate ***
Straight 30-m sprint with a puck (s)	6.4 ± 0.4	1.1%; ± 4.7%	Trivial *	−5.4%; ± 2.5%	large ****
Backward skating for 30-m without a puck (s)	7.8 ± 0.9	0.9%; ± 2.3%	Trivial *	−2.4%; ± 2.7%	small*
Straight 4-m sprint speed without a puck (s)	1.9 ± 0.1	−11.7%; ± 2.4%	Large ****	−1.7%; ± 3.8%	unclear
Straight 4-m sprint speed with a puck (s)	2.0 ± 0.1	−16.3%; ± 2.5%	very large ****	1.3%; ± 2.9%	unclear
Backward 4-m sprint speed without a puck (s)	2.4 ± 0.1	−15.2%; ± 3.4%	very large ****	1.1%; ± 4.8%	unclear

COD = change of direction training, CL = confidence limit with 90% confidence interval for the mean. Observed changes are expressed as percentages; baseline values are expressed in measurement units. Negative values indicate beneficial effects with shortened test execution times. * possible, ** likely *** very likely **** most likely, determined effect was <0.20, trivial; 0.20–0.59, small; 0.60–1.19, moderate; 1.20–1.99, large; 2.00–3.99 very large.

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
