# Peer review of "The Specificity of Motor Learning Tasks Determines the Kind of Skating Skill Development in Older School-Age Children"

_sports, 2020, doi:10.3390/sports8090126_

Round 1

Reviewer 1 Report

The article is an interesting study on the effects of certain hockey training drills on real-time speed and other hockey skills. The article would contribute to the literature, and I would recommend publication with modest changes that are listed below: 

  1. In the abstract, line 14, please hyphenate school-aged children.
  2. Line 15, change-of-direction speed-exercises should be hyphenated.
  3. Using the term "most likely" to describe improvement is odd to me.  I would like for the authors to take a stand that either statistically the treatment affected the outcome or it did not. I shall leave this up to the editor, but I would prefer more straightforward wording.
  4. Line 50, perhaps the authors should note that "generally" athletes undergo dramatic changes in body size, but it is not always the case. Perhaps the authors can finesse this sentence a bit. 
  5. Line 56, I prefer the use of the serial comma, but this is an editorial decision on formatting.
  6. Line 58, if the authors use i.e., which is the abbreviation for id est, or "that is," then the following items are a complete listing and et cetera (etc.) should not be used.  If the authors only want to have some examples, then e.g., which is the abbreviation for exempli gratia, or "for example," should be used, and, again, no et cetera (etc.) is needed as it is understood to be an incomplete listing.
  7. Line 67, the (n=13) is not needed because the sentence begins with "Thirteen."
  8. Line 75, could the authors describe the "warm-up" in a bit more detail?  It is not a big deal, but I think describing its length and the type of activities involved would make the study clearer and more reproducible.
  9. Line 85, can the authors list age, height, weight, and BMI here? Weight is missing, but it has to be known because BMI has been calculated.
  10. Line 111 is a bit confusing to me.  Why is the etc. listed? Perhaps the authors can reword this sentence to make it clearer.
  11. Line 121, the word player should be followed with his instead of their, as it is singular. 
  12. Line 158, Can the image be of better quality?  The one I am seeing is blurry, but perhaps it was damaged in the download. 
  13. Line 185, using "might be" is not the best words to use. I would prefer the authors to take a hard stand one way or the other based on the data of the study.
  14. Lines 186 and 187, use COD instead of writing Change of direction. It was introduced far earlier in the paper, so using the acronym is appropriate for the remainder or the paper.
  15. Line 195, I am not sure the age of the subject is the true defining variable here.  Perhaps the authors mean the training level is at a point where the motor skills are automated. An untrained and inexperienced 13-year-old is not going to skate like another 13-year-old who has skated regularly since he was 5 years old. Training is the key, not age, I believe.  Perhaps the authors could finesse this a bit. 
  16. Line 197, should skating-skill development be hyphenated? I think so because skating skill is modifying development. 
  17. Line 207, another reference is made to age, but I believe the authors really mean experience.  However, age can have a factor, but it would technically be maturation instead of pure age. This is a bit picky, and I do apologize to the authors, but I think being precise would make the manuscript stronger. Perhaps the authors can clarify this section a bit. 
  18. Lines 210ish to 215, could the authors describe the ages and sizes of the athletes in the various categories?  Are the PeeWee athletes grouped by age and size, or just age?  I am not familiar with these categories, but I think the paper would be stronger if the authors would describe the typical ages, sizes, etc. of the categories. Perhaps even cite a source for this or even the rulebook, if age and weight makes a difference. I do know not all 13-year-old boys develop at the same rate, and some classifications, such as PeeWee or Little Leagues or what have you, have size restrictions along with age restrictions for fairness. This may not be the case in this instance, but stating a sentence or two about this would be helpful.  
  19. Line 225, "older college players" may need to be more defined. I think it would make the paper stronger if this were quantified and qualified.
  20. Lines 237 and 243, COD again instead of change of direction. 

Those are the minor issues that I see.  If the authors address these, per the editor's satisfaction, then I am okay with acceptance of the paper.

Author Response

Reviewer 1

The article is an interesting study on the effects of certain hockey training drills on real-time speed and other hockey skills. The article would contribute to the literature, and I would recommend publication with modest changes that are listed below: 

Answer: Thank you for your detailed review, we thing that we resolved all of your valuable comments.

  1. In the abstract, line 14, please hyphenate school-aged children.

Answer: Done

  1. Line 15, change-of-direction speed-exercises should be hyphenated.

Answer: Done

  1. Using the term "most likely" to describe improvement is odd to me.  I would like for the authors to take a stand that either statistically the treatment affected the outcome or it did not. I shall leave this up to the editor, but I would prefer more straightforward wording.

Answer: Thank you for this point. We agree with more straight forward expression of results, therefore we put the designation of effect size and explicitly state the percentage change. We believe that this is much better for abstract.

  1. Line 50, perhaps the authors should note that "generally" athletes undergo dramatic changes in body size, but it is not always the case. Perhaps the authors can finesse this sentence a bit. 

Answer: We now used the term “anthropometric changes“. In terms of some individuals with late grow spurt, we stated that those changes are typical not exactly given.

  1. Line 56, I prefer the use of the serial comma, but this is an editorial decision on formatting.

Answer: We leave this decision on editorial team.

  1. Line 58, if the authors use i.e., which is the abbreviation for id est, or "that is," then the following items are a complete listing and et cetera (etc.) should not be used.  If the authors only want to have some examples, then e.g., which is the abbreviation for exempli gratia, or "for example," should be used, and, again, no et cetera (etc.) is needed as it is understood to be an incomplete listing.

Answer: We want mean just examples therefore we used e.g., only

  1. Line 67, the (n=13) is not needed because the sentence begins with "Thirteen."

Answer: Corrected

  1. Line 75, could the authors describe the "warm-up" in a bit more detail?  It is not a big deal, but I think describing its length and the type of activities involved would make the study clearer and more reproducible.

Answer: We have specified the warm up on the ice in bit more details.

  1. Line 85, can the authors list age, height, weight, and BMI here? Weight is missing, but it has to be known because BMI has been calculated.

Answer: Sorry for this confusion, we now clearly stating the body weight and BMI (18.14 ± 2.08)

  1. Line 111 is a bit confusing to me.  Why is the etc. listed? Perhaps the authors can reword this sentence to make it clearer.

Answer: Thank you for this point, we already changed and reword this sentence.

  1. Line 121, the word player should be followed with his instead of their, as it is singular. 

Answer: Corrected

  1. Line 158, Can the image be of better quality?  The one I am seeing is blurry, but perhaps it was damaged in the download. 

Answer: We check the figure quality and it is clear, however we uploaded this figure separately in high resolution now.

  1. Line 185, using "might be" is not the best words to use. I would prefer the authors to take a hard stand one way or the other based on the data of the study.

Answer: We now state that they “are beneficial”.

  1. Lines 186 and 187, use COD instead of writing Change of direction. It was introduced far earlier in the paper, so using the acronym is appropriate for the remainder or the paper.

Answer: Done

  1. Line 195, I am not sure the age of the subject is the true defining variable here.  Perhaps the authors mean the training level is at a point where the motor skills are automated. An untrained and inexperienced 13-year-old is not going to skate like another 13-year-old who has skated regularly since he was 5 years old. Training is the key, not age, I believe.  Perhaps the authors could finesse this a bit. 

Answer: Yes, training level is the correct description, thus we change this sentence according to your suggestion.

  1. Line 197, should skating-skill development be hyphenated? I think so because skating skill is modifying development. 

Answer: We agree and thus we hyphenate the skating-skills in whole article including the title.

  1. Line 207, another reference is made to age, but I believe the authors really mean experience.  However, age can have a factor, but it would technically be maturation instead of pure age. This is a bit picky, and I do apologize to the authors, but I think being precise would make the manuscript stronger. Perhaps the authors can clarify this section a bit. 

Answer: This is actually really relevant point, we now added that other studies used older and more experienced players.

  1. Lines 210ish to 215, could the authors describe the ages and sizes of the athletes in the various categories?  Are the PeeWee athletes grouped by age and size, or just age?  I am not familiar with these categories, but I think the paper would be stronger if the authors would describe the typical ages, sizes, etc. of the categories. Perhaps even cite a source for this or even the rulebook, if age and weight makes a difference. I do know not all 13-year-old boys develop at the same rate, and some classifications, such as PeeWee or Little Leagues or what have you, have size restrictions along with age restrictions for fairness. This may not be the case in this instance, but stating a sentence or two about this would be helpful.  

Answer: Thank you for this point, we have added the age categories ranges in the bracket and included reference for this categorization.

  1. Line 225, "older college players" may need to be more defined. I think it would make the paper stronger if this were quantified and qualified.

Answer: We have added clear categorization which we are following in discussion and even we specifying the age and population from different studies.

  1. Lines 237 and 243, COD again instead of change of direction. 

Answer: Done

Those are the minor issues that I see.  If the authors address these, per the editor's satisfaction, then I am okay with acceptance of the paper.

Answer: We believe that we addressed all the concerns and thank you again for improving our manuscript.

Reviewer 2 Report

The present study was designed to compare the effects of training programs using change of direction (COD) speed exercises and partial skating task (SeqT) training on speed and agility performance in U12 ice-hockey players. The authors concluded that COD training on the ice improves short starts and agility with a puck, while SeqT target longer sprints and agility without a puck. Therefore, both types of training should be applied in accordance with motor learning tasks specific to current training needs. The manuscript is well written and present interesting findings based on different training protocols. However, some modifications are required:

Introduction:

  • How did the authors hypothesize that COD training is more beneficial than SeqT. This should be supported by published studies.

Material and Methods:

  • How did the authors decide on the duration of the training (change of direction speed training) for the participants? Are they similar studies with the same duration that could be cited by the authors?
  • What is the gender of the participants in the study? Are there any studies showing different results in male and female athletes at this level?
  • Concerning the number of the participants, can the authors provide some justification that the sample size is adequate to detect differences in training method in the form of a power analysis?
  • The sentence starting at line 77 and finishing at line 78 does not seem to be completed.
  • The sentence at line 85 does not seem to be completed by the authors.
  • 2.6.3 and 2.6.4 can be combined together (with and without puck) to avoid repletion.
  • 2.6.6 and 2.6.7 can be combined together too to avoid repletion as above.
  • I would suggest to remove the texts from Figure 2 and only keep test numbers in the figure.

Results:

  • Out of 13 participants, the authors mentioned that they were both forwards and defenders. Were they any variability in the results based on them being forwards or defenders?

Discussion:

  • I would suggest adding a limitation part.

Author Response

The present study was designed to compare the effects of training programs using change of direction (COD) speed exercises and partial skating task (SeqT) training on speed and agility performance in U12 ice-hockey players. The authors concluded that COD training on the ice improves short starts and agility with a puck, while SeqT target longer sprints and agility without a puck. Therefore, both types of training should be applied in accordance with motor learning tasks specific to current training needs. The manuscript is well written and present interesting findings based on different training protocols. However, some modifications are required:

Answer: Thank you for you revisions and suggestion we believe that we resolved all of your concerns.

Introduction:

  • How did the authors hypothesize that COD training is more beneficial than SeqT. This should be supported by published studies.

Answer: We added the statement that COD has been reported for development of skills even in elite athletes and SeqT not. We added one review reference to support the COD statement.

Material and Methods:

  • How did the authors decide on the duration of the training (change of direction speed training) for the participants? Are they similar studies with the same duration that could be cited by the authors?

Answer: We followed previous research performed on ice-hockey players, which we are citing in the method section now.

  • What is the gender of the participants in the study? Are there any studies showing different results in male and female athletes at this level?

Answer: They were males which is now stated in method section. There are reported differences between males and females, but in the tests which are different then ours. We have added the statement about those differences in limitation section.

  • Concerning the number of the participants, can the authors provide some justification that the sample size is adequate to detect differences in training method in the form of a power analysis?

Answer: According to Hopkins 2009, we have used appropriate sample size for intervention and clinical interpretation which is more than 10, when using inference results. We already citing this reference below in statistic section for the justification the inference results, which we believe is enough to state.

Hopkins, W., Marshall, S., Batterham, A., & Hanin, J. (2009). Progressive statistics for studies in sports medicine and exercise science. Medicine+ Science in Sports+ Exercise41(1), 3.

We calculated minimum sample size for one group pre-post design, which is in our case 90% CI, 80% power was n =12. However, since we are not presenting the results of T test we did not added this calculation to the statistical section.

  • The sentence starting at line 77 and finishing at line 78 does not seem to be completed.

Answer: Improved, the sentence if now corrected.

  • The sentence at line 85 does not seem to be completed by the authors.

Answer: Thank you for this typo, we completed this sentence to flow with following text.

  • 6.3 and 2.6.4 can be combined together (with and without puck) to avoid repletion.

Answer: We combined actually three test into one paragraph to avoid even more repetitions. In accordance with this recommendation we combined also previous agility tests.

  • 6.6 and 2.6.7 can be combined together too to avoid repletion as above.

Answer: We combined actually three test into one paragraph to avoid even more repetitions.

  • I would suggest to remove the texts from Figure 2 and only keep test numbers in the figure.

Answer: Thank you for this suggestion, we moved the test descriptions and line marks out of the figure.

Results:

  • Out of 13 participants, the authors mentioned that they were both forwards and defenders. Were they any variability in the results based on them being forwards or defenders?

Answer: Because of small number of participants we put all participants (forwards and defenders) together. The reason is also that the current requirements for sprinting and agility are quite similar since all players are required to performed offense and defense skills individual tactics and switch their playing positions during a match.

Discussion:

  • I would suggest adding a limitation part.

Answer: We agree, and we have added the limitation section

Reviewer 3 Report

Add year of the study

Units table 1

Add table with baseline characteristics of the sample

How can have an effect the fact that they performed first exercises for directional speed and after partial skating exercises? Can these produce bias? 

Add the adherence to the trainings

Any injury?

Who established (like exercise physiology or trainers) the tests and trainers? add this information

Add strength and limitations of the study

Discussion is quite short. 

Author Response

Reviewer 3

Add year of the study

Answer: Done in methods

Units table 1

Answer: All test were in seconds, which is now stated in the table.

Add table with baseline characteristics of the sample

Answer: This table has been added in subject section and test baseline values are in the table 2.

How can have an effect the fact that they performed first exercises for directional speed and after partial skating exercises? Can these produce bias? 

Answer: We agree that missing cross-sectional design might favor the first COD intervention in our case. However, since the players in the study are interchanging both types of trainings during regular season, we can neglect large learning effect from the first intervention. Moreover, we can highlight the specificity of motor learning task, which can specify the use of both methods when coaches observe deficits in agility or straight speed.

In relation to your comment we have added this fact into limitation section.

Add the adherence to the trainings

Answer: All participants underwent all training sessions, which we added into the result section. There was more players during the first testing session, however we did not included participants, who were not able to fully participate in the study.

Any injury?

Answer: During this study we did not have any injury in research group, which we added into the result section

Who established (like exercise physiology or trainers) the tests and trainers? add this information.

Answer:

The training intervention was established by head coach AT in cooperation with research team DM, PS, PL containing exercise physiologist, biomechanics and sport science experts. Which is now stated in method section. During experiment we control individual warm-up, Performed exercises, tests and whole training program.

Add strength and limitations of the study

Answer: We agree, and we have added the limitation section and highlight the study strength.  

Discussion is quite short. 

Answer: The discussion has been enlarged by limitation section and some relation to the test results.